# Current Biological, Pathological and Clinical Landscape of HER2-Low Breast Cancer

**DOI:** 10.3390/cancers15010126

**Published:** 2022-12-25

**Authors:** Huina Zhang, Yan Peng

**Affiliations:** 1Department of Pathology, University of Rochester Medical Center, Rochester, NY 14642, USA; 2Department of Pathology and Simmons Comprehensive Cancer Center, University of Texas Southwestern Medical Center, Dallas, TX 75390, USA

**Keywords:** HER2, breast cancer, HER2-low, antibody-drug conjugate, trastuzumab-deruxtecan, T-Dxd

## Abstract

**Simple Summary:**

The breakthrough in developing novel HER2-targeting antibody drug conjugates and identifying their clinical benefits in HER2-low breast cancer will dramatically revolutionize the clinical treatment landscape of HER2 negative breast cancers, as well as the pathologic evaluation of HER2 status in breast cancers. This review updates the current biological, pathological and clinical landscape of HER2-low breast cancer and proposes the future directions on clinical management, pathology practice, and translational research in this subset of breast cancer.

**Abstract:**

HER2-low breast cancer (BC) is a newly defined subset of HER2-negative BC that has HER2 immunohistochemical (IHC) score of 1+ or score of 2+/in situ hybridization (ISH) negative phenotype. Recent clinical trials have demonstrated significant clinical benefits of novel HER2 directing antibody-drug conjugates (ADCs) in treating this group of tumors. Trastuzumab-deruxtecan (T-Dxd), a HER2-directing ADC was recently approved by the U.S. Food and Drug Administration as the first targeted therapy to treat HER2-low BC. However, HER2-low BC is still not well characterized clinically and pathologically. This review aims to update the current biological, pathological and clinical landscape of HER2-low BC based on the English literature published in the past two years and to propose the future directions on clinical management, pathology practice, and translational research in this subset of BC. We hope it would help better understand the tumor biology of HER2-low BC and the current efforts for identifying and treating this newly recognized targetable group of BC.

## 1. The Natural History of HER2-Low Breast Cancer

Human epidermal growth factor receptor 2 (HER2) is an important prognostic and predicative biomarker in breast cancer (BC) and all patients with newly diagnosed primary or metastatic BCs should be tested for HER2 protein expression by immunohistochemistry (IHC) and/or gene expression by in situ hybridization (ISH) to guide the clinical management [1,2,3]. BCs are currently classified as HER2 positive when HER2 expression is scored 3+ by IHC or an IHC score of 2+ with HER2 gene amplification tested by ISH. Patients with HER2-positive tumors are eligible for HER2-pathway blockade agents including anti-HER2 monoclonal antibodies (trastuzumab, pertuzumab and margetuximab), anti-HER2 antibody drug conjugates (ADCs) [trastuzumab emtansine (T-DM1) and trastuzumab deruxtecan (T-Dxd)] and tyrosine kinase inhibitors (tucatinib, lapatinib and neratinib). These HER2-targeted agents have dramatically improved the clinical outcomes of HER2-positive BCs [4]. On the contrary, BCs with HER2 IHC score of 0 or 1+, or an IHC score of 2+ without gene amplification are considered as HER2-negative and these tumors lack a significant clinical benefit from these traditional HER2-pathway blockade agents.

Among HER2-negative BCs, hormonal receptor (HR)-negative tumor (triple negative, TNBC) is a biologically aggressive subtype of BC with a poor prognosis and limited treatment options. Currently, the mainstay treatment for TNBCs is cytotoxic chemotherapy, although other therapies including immunotherapy are expanding [5,6]. In a recent meta-analysis, Li et al. reported median overall survival (OS) of 17.5 and 8.1 months and median progression-free survival (PFS) of 5.4 months and 2.9 months in patients with metastatic TNBCs after first-line chemotherapy and later lines of treatment, respectively [7]. In addition, although HR positive, HER2-negative BCs have overall relatively favorable prognosis, most advanced/metastatic HR positive/HER2 negative BCs remain incurable with a median overall survival of 24.8 months [8]. Up to 40% of advanced/metastatic HR positive/HER2-negative BCs respond to the first line treatment, but eventually develop endocrine therapy resistance and options are limited for later line of therapy [8]. The limited activity and associated unfavorable toxicity profiles of chemotherapies in treating high risk or advanced HER2 negative BCs highlight a considerable unmet need for improved therapeutic options.

Promising results of recent clinical trials opened the door for treating a subset of HER2 negative BCs with HER2-targting ADCs. Banerji et al. first published a phase I clinical trial results of trastuzumab duocarmazine, a new HER2-targeting ADC in patients who had advanced BCs with HER2 IHC scores of 1+ or 2+/negative ISH. Treatment with trastuzumab duocarmazine achieved objective response in 28% (9/32) of HR positive tumors and in 40% (6/16) of patients with HR negative tumors [9]. Modi et al. subsequently reported the results of another phase Ib clinical trial which showed patients with advanced BCs and HER2 IHC scores of 1+ or 2+/ISH negative results achieved an objective response rate (ORR) of 37% after the administration of T-Dxd, a previously FDA-approved HER2-targeting ADC for metastatic HER2 positive BCs [10]. Based on these promising clinical trial results, in 2020, Tarantino et al. first proposed the concept of “HER2 low” in BC which refers to BC with HER2 IHC score of 1+ or 2+/ISH negative result [11]. Figure 1 illustrates the changes of HER2 scoring in BC from the current two-tier to three-tier scoring system with the addition of HER2-low category.

The result of phase 3 clinical trial (DESTINY Breast-04, DB-04) of T-Dxd in previously heavily-treated HER2-low advanced BC was published in June 2022 and it showed T-Dxd significantly improved survival in patients with HER2-low advanced BC, compared to chemotherapy of physician choice [12]. In August 2022, the U.S. Food and Drug Administration (FDA) approved T-Dxd as the first targeted therapy for the treatment of patients with unresectable or metastatic HER2-low BC [13]. The breakthrough in developing novel HER2-targeting ADCs and identifying their clinical benefits in HER2-low BC will dramatically revolutionize the clinical treatment landscape of BC, as well as the pathologic evaluation of HER2 status in BC. Figure 2 lists the key publications and timeline in the natural history of HER2-low BC.

## 2. Current Biological Landscape of HER2-Low Breast Cancer

### 2.1. The Incidence of HER2-Low Breast Cancer

HER2-low BC is estimated to account for approximately 45–55% of BCs; however, this estimation is based on studies using variable HER2 scoring criteria [11,14,15]. After the introduction of “HER2-low” in BC in 2020, few studies have reported the incidence of HER2-low BC was between 31% and 51% [16,17,18,19]. HER2-low BC is more common in HR+ positive BCs (ranges: 43.5–67.6%) than TNBCs (ranges:15.7–53.6%) [20]. More specifically, in the advanced BCs, the reported incidence of HER2-low BC ranged from 35.2–63.2% [19,21,22].

### 2.2. The Biology of HER2-Low Breast Cancer

Currently, the knowledge on the biology of HER2-low BC is still limited and it appears to represent a group of breast tumors with significant biological heterogeneity. Both pooled-analysis of large cohorts and smaller single-institutional studies have revealed that most of HER2-low tumors are luminal molecular subtypes (Luminal A: 29.3–65.5%; Luminal B: 22.8–50.5%; HER2-enriched: 1.1–4.1%; Basal: 4.6–7.7%) [16,18,23,24], are enriched in HR positive BCs [16,19,25,26,27,28,29,30,31,32], have a lower Ki-67 proliferation index [17,18,25,26,27,32], and are less responsive to neoadjuvant chemotherapy (NAC) with a pCR rate between 9.8% and 36.3% [17,25,26,32,33,34,35,36,37,38].

Whether HER2-low BC represents a distinct biologic/clinical group and the HER2-low expression has prognostic significance in BC, especially when compared to HER2-0 BC (BC with HER2 IHC score of 0), remain controversial in the current literature [17,26,28,31,33,39,40]. In a large cohort study by Denkert et al., HER2-low BC appeared to be a distinct biological subtype in HER2-negative BCs because this group of tumors had different clinicopathologic characteristics, less responsive to NAC, and relatively-better survival in therapy-resistant HR-negative BCs [17]. On the contrary, in another large cohort study of 5235 patients with HER2-negative BCs, Tarantino et al. [31] found that most of clinicopathologic differences between HER2-0 and HER2-low BCs were associated with HR status. Compared to HER2-0 BC, HER2-low BC had no prognostic significance when adjusting to HR status. The results of that study failed to support the interpretation of HER2-low as a distinct biologic subtype of BC [31]. Likewise, the superior prognosis in HER2-low than HER2-0 tumors has been reported in several studies [17,28,30,41]; while, other studies have failed to demonstrate any prognostic values of HER2-low status in BCs [16,19,29,31,38,39,42,43,44]. Based on these conflicting results, no clear conclusions can be drawn at present time on both the prognostic value of HER2-low expression and the distinct biologic/clinical entity of HER2-low BC, and this is likely due to differences in the studied patient population, the HR status, the study design/endpoint, and/or follow-up duration. More studies, especially prospective studies which include HR status and treatment protocols may be helpful to better understand the biology of this group of BCs.

The dynamic change associated with HER2-low BC in both primary BCs and matched local recurrences/distant metastases or post-NAC tumors has also been reported. Both Miglietta et al. [22] and Tarantino et al. [45] reported that HER2-low expression was highly unstable during disease progression, and there was a significant discordance (38% and 66%) in HER2-low expression between primary tumors and matched advanced stage tumors, with enrichment of HER2-low carcinomas in the advanced setting. It was also demonstrated that 26.4% of patients had discordant HER2 expression between pre- and post-NAC treatments, mostly seen in cases converting either from HER2-low (14.8%) or to HER2-low (8.9%) expression [46].

### 2.3. The Molecular Basis of HER2-Low Breast Cancer

Compared to HER2-0 BC, HER2-low BC has been reported to have a higher *ERBB2* mRNA expression [16,47], increased prevalence of *PIK3CA* mutation [17,26,48] and reduced *TP53* mutation [17,48]. A higher prevalence of *FGFR1* amplification (defined as ≥10 copy number gain) in the HER2-low group (12% vs. 1.8%) compared to HER2-0 carcinomas was reported [48].

To gain insights into the molecular basis of HER2-low BC, Berrino et al. performed high-throughput molecular analysis on 99 HER2-low BC tissue samples and compared the mutation rates and gene expression profiles of HER2-low BC with those of HER2-negative and HER2-positive BCs in a Memorial Sloan-Kettering Cancer Center BC cohort [24]. The results showed that the most common mutations in HER2-low BC were *PIK3CA* (31/99, 31%), *GATA3* (18/99, 18%), *TP53* (17/99, 17%), and *ERBB2* (8/99, 8%). In addition, the RNA-based class discovery analysis also unveiled four subsets in the HER2-low tumors using LAURA classification: 1) lymphocyte activation, 2) unique enrichment in HER2-related features, 3) stromal remodeling alterations, and 4) actionability of *PIK3CA* mutations. Tumor mutational burden was significantly higher in HER2-low BC with IHC score 1+ compared to those with IHC score 2+, HER2/CEP17 ratio < 2 and copy number between 4 and 6 (*p* = 0.04). Comparison of mutation spectra revealed that HER2-low BC was different from both HER2-0 and HER2-positive BCs, with score 1+ tumors resembling more the HER2-0 tumors and score 2+/ISH negative tumors more related to the HER2-positive tumors. Intra-group gene expressions also demonstrated overlapping features between IHC score 1+ tumors and HER2-0 BCs, whereas tumors with IHC score 2+, HER2 HER2/CEP17 ratio <2 and copy number between 4 and 6 showed the highest diversity [24]. van den Ende et al. studied the gene expressions in 429 HER2-0 and 100 HER2-low BCs. HER2-low tumors were found to have higher Era Like 12S Mitochondrial RRNA Chaperone 1 (*ERAL1*), Mediator Complex Subunit 24 (*MED24*) and Post-GPI Attachment to Proteins Phospholipase 3 (*PGAP3*) gene expression, likely due to the amplification of a common chromosomal region. In addition, HER2-low BC was associated with a limited immune response compared to HER2-0 BC, as demonstrated by the gene-expression data in the ER-positive tumors and the tumor infiltrating lymphocytes-score in the ER-negative cohort [47]. 

### 2.4. Factors may Contribute to the HER2-Low Expression in Breast Cancer

Although BCs have been scored as HER2 0, 1+, 2+ or 3+ by IHC, the numbers of HER2 receptor molecule in human breast cancer cells are continuously distributed, ranging from approximately 20,000 per cell in normal breast epithelium and HER2 IHC 0 BCs to approximately 2,300,000 per cell in HER2 IHC 3+ BCs [49]. Several factors have been speculated to contribute to the HER2-low expression in BC including the bi-directional crosstalk between ER and HER2, the modification effect by endocrine therapy, and the activation of NF-*k*B pathway by chemotherapy and radiation therapy [11]. It is well-documented that the presence of complex bi-directional molecular crosstalk between the ER and HER2 pathways in BC plays a significant role in the development of tumor resistance to endocrine or HER2-targeted therapies, since treatment strategies targeting either pathway result in the upregulation of the other one [50,51]. In the recently published literature, it has been consistently demonstrated that HER2-low BC is enriched in HR+ tumors and majority of HER2-low tumors are ER+ [16,19,25,26,27,28,29,30,31], indicating ER signaling contributes significantly to the HER2-low expression and related tumor biology. In addition, the dynamic changes of HER2 expression in HER2-low tumors between primary and metastatic/recurrent/NAC BCs further support the roles of endocrine therapy, chemotherapy and radiation therapy in shaping the HER2-low expression in BC [22,45,46].

## 3. Current Pathological Landscape of HER2-Low Breast Cancer

### 3.1. The Fundamental Challenge in the Pathological Landscape of HER2-Low Breast Cancer: Accurate Definition

Due to lack of clinical benefits from the HER2-pathway blockade agents, HER2-low expressing BC has long been disregarded as an epiphenomenon without clinical implication of considering HER2-targted therapy. The American Society of Clinical Oncology (ASCO)/College of American Pathologists (CAP) HER2 testing guidelines in BC, initially published in 2007, and updated in 2013 and 2018, have been focusing on separating patients with HER2 positive tumors who are eligible for HER2-targeted therapy, from those with HER2 negative tumors [1,2,3].

There is no formal definition for HER2-low BC up to this point and this is a fundamental challenge in the current pathological landscape of this newly recognized, targetable tumor group, especially for developing an accurate testing method. The widely used HER2 IHC 1+ or 2+ with negative ISH result for defining HER2-low in BC is based on the inclusion criteria of those clinical trials. In the DB-04 study, the PFS benefit of T-Dxd was consistently observed in patients with HER2 IHC 1+ (10.3 months) and IHC 2+/ISH-negative disease (10.1 months) [12], indicating the number of HER2 receptor molecules in the IHC 1+ tumor cell membrane (~100,000 molecules) reached the threshold for the unique bystander killing effect of T-Dxd in HER2-low expressing tumors. However, the current definition for HER2-low BC may not be an adequate representation of the target population for these novel ADCs and how to decide the lower end of HER2 expression to define HER2-low BC is still evolving. Preliminary results from a phase II clinical trial demonstrated similar response rates to T-Dxd treatment between advanced BC patients with HER2 IHC score of 0+ (30.6%) and HER2-low (33.3%) [52]. In the ongoing DB-06 trial (NCT04494425), which is designed to evaluate the efficacy of T-Dxd in metastatic HER2-low/HR-positive metastatic BC patients with disease progression on endocrine therapy, BC patients with both HER2-low and HER2 IHC expression of >0 and <1+ (currently considered as HER2-0, or ultra-low) are included. Hopefully the results of this trial will provide more evidence on defining the threshold levels /lower limits of HER2 expression required to benefit from an ADC therapeutic approach, such as T-Dxd, and determining the clinical significance of distinguishing HER2-low BC from HER2-0 tumor by using current testing methods.

### 3.2. IHC Testing as the Primary Method for Identifying HER2-Low Breast Cancer

Under the current definition, the identification of HER2-low BC relies on the IHC testing protocol and scoring system as laid out in the ASCO/CAP guidelines [1,2,3]. In the DB-04 study, the VENTANA anti-HER2/neu (4B5) IUO Assay system (with ISH testing when applicable) was used to identify patients with HER2-low status, which suggested that a conventional IHC test can accurately identify patients who may benefit from T-Dxd [12]. However, as a semi-quantitative test, HER2 IHC testing was primarily developed to help separate high levels of HER2 expression [~2 millions of molecules per cell, corresponding to HER2 IHC score 3+] from lower level HER2 expression (~20,000–500,000 molecules per cell, corresponding to HER2 IHC 0–2+) and may not be an ideal method for detecting HER2-low BC. The pre-analytic, analytic and post-analytic variables of IHC testing also significantly affect the interpretation of HER2 status. We have previously emphasized the limitations and challenges of IHC as the primary testing method for identifying HER2-low BC in detail, including the notable inter-observer and inter-antibody variability in HER2 IHC scoring, especially in the HER2-low expressing tumors [15,53,54]. Since it was used in the DB-04 trial, the U.S. FDA recently approved the VENTANA PATHWAY anti-HER2/neu (4B5) Rabbit Monoclonal Primary Antibody as the first companion diagnostic test to identify metastatic BC patients with low HER2 expression for whom T-Dxd may be considered as a targeted treatment [55]. This approval would have a significant impact on the pathology practice, especially in those pathology laboratories which do not have the VENTANA platform. In addition, although it has demonstrated that HER2-low BC is targetable group of tumor in the clinical trials, those clinical trials do not necessarily serve as the platform for validating the assay of the vendor used in the trial [56]. Further efforts to address whether other commonly used, FDA-approved HER2 IHC testing methods could reliably identify appropriate patients for T-Dxd treatment are needed.

### 3.3. Current Developments in More Accurate and Reliable Methods for Identifying HER2-Low Expressing Tumors

Due to the limitations of IHC method as the primary testing for identifying patients with HER2-low tumors, more accurate and reproducible methods are urgently needed for an accurate prediction of efficacy of the novel HER2-targeting ADCs used to treat HER2-low BC. The scientific community and industry are currently making efforts to develop more accurate and reliable methods to facilitate the identification of patients with HER2-low BC.

After investigating 363 BCs with HER2 IHC scores 0, 1+ and 2+ without HER2 gene amplification, with the aid of an artificial neural network model and the correlated HER2 mRNA levels, Atallah et al. proposed a refined HER2-low definition in BC [57]. More specifically, the proposed definition for HER2 IHC score 1+ in this study was membranous staining in invasive tumor cells as either (1) faint intensity in ≥20% of cells regardless the circumferential completeness; (2) weak complete staining in ≤10%; (3) weak incomplete staining in >10% or (4) moderate incomplete staining in ≤10%. It has showed that this refined definition reached high intra-observer agreement (kappa value of 0.8) and inter-observer agreement (kappa value of 0.9) [57].

Deep learning-based technology plays an increasingly significant role in the pathology field, especially in the image analysis and quantitative evaluations. With the assistance of artificial intelligence (AI), it is feasible to develop computer algorithms to analyze HER2 IHC images and provide more objective, reproducible scoring results [58,59,60,61,62]. Gustavson et al. used deep learning-based image analysis and generated a novel HER2 Quantitative Continuous Score (QCS). The HER2 QCS was largely consistent with pathologist’s HER2 scoring, and could potentially enhance prediction of patient outcome with T-Dxd by increasing sensitivity and specificity of response, especially in the HER2-low population [63]. A study from China accessed the role of AI in the accurate interpretation of HER2 IHC 0 and 1+ in 246 BCs and found the interpretation accuracy was significantly increased with AI assistance (Accuracy 0.93 vs. 0.80), as well as the evaluation precision of HER2 0 and 1+. The AI algorithm also improved the total consistency (ICC = 0.542 to 0.812), especially in HER2 1+ cases, as well as the accuracy in cases with heterogeneity (Accuracy 0.68 to 0.89) [64]. Recently, PAIGE, an AI-based company announced CE-IVD and UKCA designations of its new digital biomarker assay, HER2Complete, to identify patients with HER2-low BC, but this software has not been approved for diagnostic procedure in the U.S. [65].

Developing accurate and quantitative methods to facilitate the identification of HER2-low BC is also under active investigation. Moutafi et al. recently developed a quantitative immunofluorescence coupled with a mass spectrometry standardized HER2 array to measure absolute amounts of HER2 protein on conventional histology sections. It showed the assay was linear between 2–20 attomols/mm^2^ which was within the range of expression in normal breast epithelium, but below the levels seen in the HER2 amplified cell lines or tumors, indicating it may allow for objective and quantitative low HER2 assessment [66]. Kennedy et al. also reported that an immunoaffinity-enrichment coupled to multiple reaction monitoring-mass spectrometry (immuno-MRM-MS) had acceptable analytical characteristics, high concordance with predicate assays, even at low HER2 expression levels [67]. In addition, a study by Xu et al. suggested that molecular method such as mRNA may better serve on defining HER2-low cancer for the treatment decision needs due to its relatively broader dynamic range [68]; while, a recent study found neither IHC nor HER2 mRNA measured by qRT-PCR method would be optimal to quantify HER2-low expression, especially for HER2 1+ BC [69].

These exciting results in the quantitative measurement of HER2 protein expression have opened the doors, but efforts are largely needed in developing cost-effective and easily-implemented methods for facilitating patient selection in the HER2-low era. Furthermore, as we mentioned previously [15], any of new quantitative methods on HER2 protein detection will need to undergo extensive analytic and clinical validation to demonstrate level 1 evidence of clinical utility before approval for the use in clinical practice.

### 3.4. HER2 Evaluation and Reporting in Breast Cancer in the New HER2-Low Era

Until more accurate and reliable quantitative methods are available for identifying HER2-low BC in routine clinical practice, IHC stays as the primary method to select patients with HER2-low expressing BCs who may benefit from the newly approved HER2-targeted agent. It is important for the pathologist to be familiar with the concept and definition of HER2-low BC and be aware of any changes in the ASCO/CAP HER2 testing guidelines. When evaluating HER2 status in BC, we should adhere strictly to the most updated ASCO/CAP HER2 testing guidelines and carefully evaluate HER2 IHC slide, especially at 400×. Pursuing consensus opinion among pathologists on challenging/borderline cases, and repeating HER2 IHC on the same or different block as well as communicating with clinical team can be helpful. Additional training may be necessary for the accurate and reproducible evaluation of HER2-low phenotype. In addition, revising the pathology report to include the HER2 IHC score and the staining patterns would provide valuable information for the clinical team to decide whether patient is eligible for T-Dxd treatment.

## 4. Current Clinical Landscape of HER2-Low Breast Cancer: Role of HER2-Targeted Agents

### 4.1. Limited Activity of Anti-HER2 Monoclonal Antibodies in HER2-Low Breast Cancer

In the retrospective subgroup analysis of two landmark adjuvant trastuzumab trials (NSABP B-31 and N9831), 174 tumors in NSABP B-31 trial and 103 tumors in N9831 trial were reclassified from HER2 positive to negative after central review, and these patients appeared possibly to benefit from additional trastuzumab therapy [70,71]. However, the recent NSABP B-47/NRG oncology phase III trial demonstrated the unequivocal evidence that the addition of trastuzumab to adjuvant chemotherapy did not benefit women with high-risk HER2-low BCs [72]. With a median follow-up of 46 months, the addition of trastuzumab to the standard adjuvant chemotherapy did not improve DFS (89.8% vs. 89.2%; hazard ratio, 0.98; *p* = 0.85), distant recurrence-free interval (92.7% vs. 93.6%; hazard ratio, 1.10; *p* = 0.55), or OS (94.8% vs. 96.3%; hazard ratio, 1.33; *p* = 0.15). Similarly, pertuzumab has limited activity in patient with HER2-low metastatic BCs, with only 4.9% (2/78) of patients achieved partial response as monotherapy, or very narrow therapeutic window with high incidence of diarrhea when combined with pertuzumab and paclitaxel [73,74]. It has been reported that margetuximab showed anti-tumor activity against HER2-low expressing cell lines in an in vitro study [75], and a phase 2 clinical trial on the activity of margetuximab in relapsed or refractory advanced BC with HER2-low expression (NCT01828021) was not completed due to lack of efficacy in 17 patients (68%), adverse effect in 2 patients (20%) and withdrawal in 1 patient (4%) [76].

### 4.2. Limited Activity of First Generation of HER2-Targeted ADC in HER2-Low Breast Cancer

ADC is a novel class of anticancer agents, which consists of a recombinant monoclonal antibody (mAbs), a cytotoxic drug (payload) and a synthetic linker. By targeting the antigen in the cell membrane by the mAb, the payload will be delivered into the targeted cells more specifically, thus improving the efficacy of payload and significantly reducing the systemic toxicities. T-DM1 is the first HER2-targeted ADC that received the U.S. FDA approval as second or beyond-line for HER2-positive BC and as adjuvant treatment for HER2 positive patients with residual disease after NAC. Currently, there is no formal clinical trial on evaluating T-DM1 in HER2-low BC. In an exploratory analysis of two phase 2 trials designed for HER2-positive BC (TDM4258 g and TDM4374 g), T-DM1 showed a lower ORR (4.8 vs. 33.8% in 4258 g, and 20 vs. 41.3% in 4374 g) and PFS (2.6 vs. 8.2 months in 4258 g, and 2.8 vs. 7.3 months in 4374 g) for HER2-negative than HER2-positive BCs [77,78].

### 4.3. Significant Clinical Benefit of New HER2-Targeted ADC in HER2-Low BCs

T-Dxd is the 2nd U.S. FDA-approved HER2-targeted ADC in metastatic HER2-positive BC and the first HER2-targeted agent in metastatic or inoperable HER2-low BC. It is composed of an anti-HER2 immunoglobulin G1 antibody, a tetrapeptide-based cleavable linker and a membrane permeable topoisomerase I inhibitor payload with a drug-to-antibody ratio of 8:1. A randomized phrase III, DB-04 trial evaluated T-Dxd in 557 patients (494 HR positive and 63 TNBCs) with HER2-low unresectable or metastatic BC previously treated with one or two lines chemotherapy [12]. Treatment with T-Dxd (5.4 mg/kg, every 3 weeks), in addition to chemotherapy of physician’s choice, resulted in a confirmed objective response rate of 52.6% in HR-positive patients and 52.3% in the overall study population compared to chemotherapy of physicians’ choice (16.3%). Compared to the chemotherapy of physician’s choices, T-Dxd significantly improved PFS in HR-positive patients (10.1 vs. 5.4 months, hazard ratio 0.51, *p* < 0.001) and in the overall population (9.9 vs. 5.1 months, hazard ratio 0.50, *p* < 0.001). OS was also improved by T-Dxd treatment among HR-positive patients (23.9 vs. 17.5 months, hazard ratio 0.64, *p* = 0.003) and the overall population (23.4 vs. 16.8 months, hazard ratio 0.64, *p* = 0.001). Similarily, in an exploratory analysis in a small number of patients with TNBCs, T-Dxd also improved PFS (8.5 vs. 2.9 months, hazard ratio 0.46) and OS (18.2 vs. 8.3 months, hazard ratio 0.48) [12].

In contrast to other anti-HER2 agents, the unique clinical benefits of T-Dxd in HER2-low BC might be achieved by the so-called “bystander killing” mechanisms due to the highly membrane-permeable payload, high drug-to-antibody ratio and cleavable linker. An in vitro study has demonstrated that T-Dxd could induce a potent “bystander killing” effect on cells in close proximity to targeted HER2-expressing tumor cells by transferring the released payload into the neighboring cells, regardless of their HER2 status [79]. It appears that in HER2-low BC, HER2 molecules on the tumor cell surface primarily function as a means for delivering antibody conjugated drugs, instead of direct inhibition of HER2 dimerization or the blockade of downstream signaling [15].

T-Dxd has generally manageable and tolerable safety profile with gastrointestinal disturbances, myelotoxicity, and alopecia being most common adverse effects. Approximately 28% of patients developed adverse reactions, and 16% of patients had to stop receiving the drug permanently during the clinical trial [80]. Interstitial lung disease (ILD)/pneumonitis is the most concerning adverse event associated with T-Dxd treatment. In a recent pooled analysis of 1150 patients who received one or more dose of ≥5.4 mg/kg T-Dxd monotherapy from nine phase I and II clinical trials, T-Dxd-related ILD/pneumonitis was found in 15.4% of patients, and most were low grade (77.4%, grade 1 or 2; 3.4% grade 3 and above) and occurred in the first 12 months of treatment [81]. Age <65 years, enrollment in Japan, T-Dxd dose >6.4 mg/kg, oxygen saturation <95%, moderate/severe renal impairment, presence of lung comorbidities, and time since initial diagnosis >4 years are the factors of interest associated with any-grade adjudicated drug-related ILD/pneumonitis [81,82]. It needs to be mentioned that with the implementation of updated guidelines for the management of toxic effects in 2019, in the DB-04 trial, the numerical incidence of high-grade events (grade 3 and above) decreased to 2.1% [12]. The specific mechanism of lung injury by T-Dxd is not clear, although it was hypothesized that the ADC-induced alveolar damage is likely due to the target-independent uptake of the conjugated payload by immune cells [82,83]. During T-Dxd treatment, if patients develop dry cough, dyspnea, fever or other new or worsening respiratory symptoms, prompt clinical and imaging evaluation of potential ILD/pneumonitis is warranted [82,84], and permanent discontinuation of T-Dxd in patients with grade 2 or higher ILD/pneumonitis should be considered. Further work is needed to better understand the pathophysiology of T-Dxd-associated ILD/pneumonitis along with the delineation of risk factors, prevention, and treatment measures, ultimately to improve the safety profile of T-Dxd for treating HER2-low BC.

### 4.4. Other Clinical Development of Agents in the Setting of HER2-Low Breast Cancer

Currently, there are several other agents under clinical development for treating HER2-low BC, including T-Dxd combined with nivolumab, T-Dxd combined with Durvalumab, trastuzumab duocarmazine, disitamab vedotin (RC-48), ARX788, A166, FS-1502 and zenocutuzumab [20].

## 5. Future Directions

Compared to 2 years ago when the concept of “HER2-low” in BC was first introduced, our understanding of HER2-low BC has significantly advanced, especially on the tumor biology. Nevertheless, there are still challenges and unanswered questions that need to be addressed, including pathology practice, translational research and clinical management in this subset of BCs. We herein propose the following future directions on the topic of HER2-low BC:What is the most appropriate/accurate assay to use for identifying HER2-low BC in the clinical practice;How to best incorporate this new classification of BC into scoring approaches and what changes are needed in terms of test implementation, validation and quality control measures;To establish a more accurate and reproducible definition of HER2-low BC;To further address whether HER2-low tumor represents a distinct clinical entity and whether HER2-low expression has prognostic value, especially prospective studies that include HR status and treatment protocols;Real-world experience with large multicenter case series on the treatment pattern and efficacy of T-Dxd in HER2-low BC;How should T-Dxd be sequenced with other treatment options for treating HER2-low BC and further evaluation of treatment combination strategies of T-Dxd with other drugs;To investigate the pathophysiology of ADC-associated adverse events especially ILD/pneumonitis along with the delineation of risk factors, prevention, and treatment measures, ultimately to improve the safety profile of these ADCs for treating HER2-low BC.

## 6. Conclusions

The development of novel HER2-targeting ADCs and identification of their significant clinical benefits in HER2-low BC, currently defined as BC with HER2 IHC 1+ or 2+/ISH negative phenotype, will dramatically revolutionize the clinical treatment landscape of HER2 negative BCs. Although our understanding of HER2-low BC has significantly advanced in the past 2 years, further efforts including basic and translational research as well as clinical studies in this newly recognized targetable group of BC are still largely needed. Figure 3 summarizes the current biological, pathological and clinical treatment landscape of HER2-low BC and our proposal for future directions on clinical management, pathology practice, and translational research in this subset of BC.

## Figures and Tables

**Figure 1 cancers-15-00126-f001:**
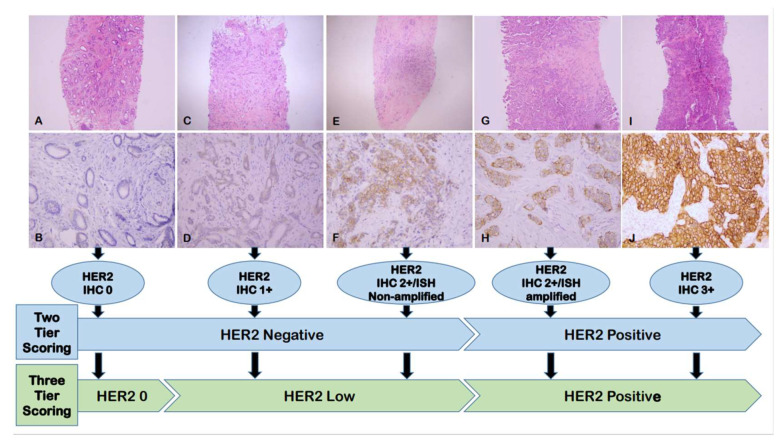
The changes of HER2 scoring in BC from the current two-tier to three-tier scoring system with the addition of HER2-low category. (**A**,**C**,**E**,**G**,**I**): Hematoxylin & eosin staining of breast cancer, ×40; (**B**,**D**,**F**,**H**,**J**): Corresponding HER2 IHC, ×200. Abbreviations: HER2: Human epidermal growth factor receptor 2; IHC: Immunohistochemistry; ISH: In situ hybridization.

**Figure 2 cancers-15-00126-f002:**
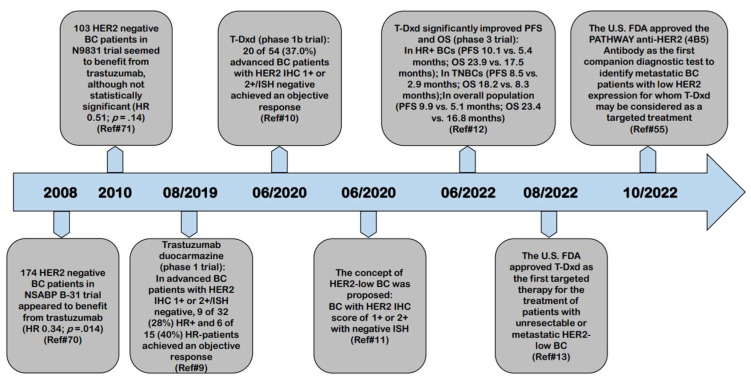
Key publications and timeline in the natural history of HER2-low BC. BC: Breast cancer; DFS: Disease-free survival; FDA: Food and Drug Administration; HER2: Human epidermal growth factor receptor 2; HR: Hazard ratio; HR: Hormonal receptor; IHC: Immunohistochemistry; ISH: In situ hybridization; OS: Overall survival; PFS: Progression-free survival; T-Dxd: Trastuzumab deruxtecan; TNBC: Triple negative breast cancer.

**Figure 3 cancers-15-00126-f003:**
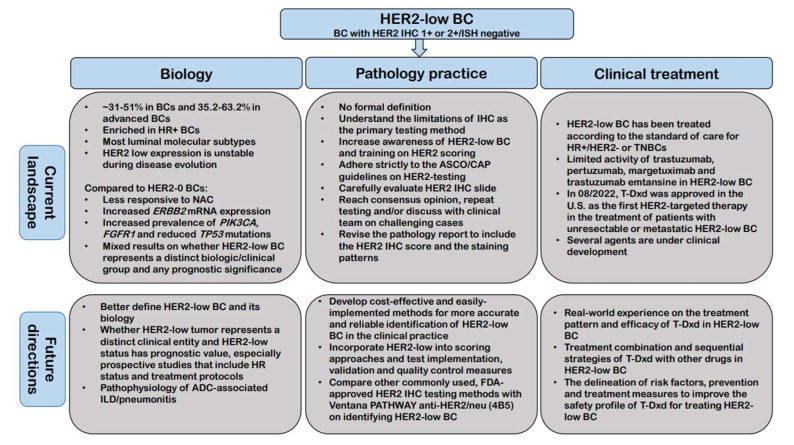
Summary of the current biological, pathological and clinical treatment landscape of HER2-low BC and our proposal for future directions on clinical management, pathology practice, and translational research in this subset of BC. Abbreviations: ADC: Antibody-drug conjugate; ASCO/CAP: The American Society of Clinical Oncology (ASCO)/College of American Pathologists (CAP); BC: Breast cancer; HER2: Human epidermal growth factor receptor 2; IHC: Immunohistochemistry; HR: Hormonal receptor; ILD: Interstitial lung disease; NAC: Neoadjuvant chemotherapy.

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
