# Peer review of "Current Biological, Pathological and Clinical Landscape of HER2-Low Breast Cancer"

_cancers, 2022, doi:10.3390/cancers15010126_

Round 1

Reviewer 1 Report

The authors reviewed recent findings regarding clinical/biological characteristics of HER2-low breast cancers. A lot of important previous clinical/clinicopathological studies are cited in the manuscript and this manuscript might help better understanding of HER2-low breast cancers.

If possible, please discuss about crosstalk between HER2 and ER.  As the authors mentioned in the manuscript, HER2-low is associated with ER positivity and the crosstalk between HER2 and ER has been well discussed in breast cancer as an important mechanism endocrine resistance.

Author Response

Answer to the reviewer's comment: This comment has been addressed by discussing the ER-HER2 crosstalk briefly and adding additional references. Please see lines 182-186 (the clean version) in the revised manuscript.

Reviewer 2 Report

The timely and informative review by Zhang et al. effectively summarizes our current knowledge of the biology and pathology of HER-low breast cancer and describes challenges of translating this information into clinical practice. It is a valuable resource for both clinicians and researchers.

Minor recommendations:

The language in several of the sentences in the simple summary and abstract are too similar and should be revised to eliminate this repetition.

Avoid passive voice phrases such as “it has called” (line 431)

Minor grammatical errors throughout the manuscript should be corrected.

Author Response

Comment #1. The language in several of the sentences in the simple summary and abstract are too similar and should be revised to eliminate this repetition.

Answer to comment #1: This comment has been addressed by revising the simple summary to reduce the repetition. Please see the Simple Summary (lines 13-16) (the clean version) in the revised manuscript.

Comment #2: Avoid passive voice phrases such as “it has called” (line 431)

Answer to comment #2: This comment has been addressed by revising the sentence in Conclusion (lines 444-447) (the clean version) in the revised manuscript.

Comment #3:  Minor grammatical errors throughout the manuscript should be corrected.

Answer to comment # 3: We read through the manuscript closely and tried to correct all the grammatical errors in this revised manuscript.